# Evaluation of Risk Factors for Fall Incidence Based on Statistical Analysis

**DOI:** 10.3390/ijerph22050748

**Published:** 2025-05-09

**Authors:** Da Hye Moon, Tae-Hoon Kim, Myoung-Nam Lim, Seon-Sook Han

**Affiliations:** 1Department of Internal Medicine, Kangwon National University Hospital, Chuncheon 24289, Republic of Korea; ansekgo@naver.com; 2Department of Internal Medicine, College of Medicine, Kangwon National University, Chuncheon 24341, Republic of Korea; blessing0104@naver.com; 3Biomedical Research Institute, Kangwon National University Hospital, Chuncheon 24289, Republic of Korea; lmn99054@kangwon.ac.kr

**Keywords:** fall, risk factor, hospital inpatient, clinical department, nursing shifts

## Abstract

Background: Falls are common among hospitalized patients, particularly affecting older adults. This study analyzed patients who experienced falls at Kangwon National University Hospital (KNUH) and classified them based on department and nursing shift hours. Methods: Data from adult patients admitted to KNUH between 2018 and 2023 who experienced falls were analyzed, focusing on demographics, medications, comorbidities, alcohol and smoking histories, and the Morse Fall Scale. The goal was to identify the key variables contributing to falls in hospitalized patients. Results: From 2018 to 2023, 336 internal medicine and 159 surgical patients experienced falls. Surgical patients had a longer length of stay (34.49 ± 47.52 vs. 24.63 ± 28.37 d, *p* = 0.016), and falls occurred more frequently during night shifts. Surgical patients had longer hospital stays (34.49 ± 47.52 vs. 24.63 ± 28.37 days), took more medications (9.20 vs. 6.83), and experienced falls sooner after narcotic use (3.77 vs. 6.17 days) than internal medicine patients. Patients who fell during night shifts were older, while those who fell during day shifts had a longer length of stay. Conclusions: The study found higher fall rates in internal medicine patients who had shorter lengths of stay and took fewer medications. Further research is needed on fall risk factors and prevention strategies.

## 1. Introduction

A fall refers to an involuntary loss of balance, resulting in a person stumbling or falling to the ground. According to a 2021 report by the World Health Organization, approximately 684,000 fatal falls occur annually, with the mortality rate from falls being highest among adults aged ≥60 years worldwide [1]. In South Korea, according to statistics from the Korea Disease Control and Prevention Agency, falls were most prevalent among the age groups 0–14 years and ≥75 years. In 2022, falls accounted for the highest proportion, of emergency room patients by mechanism of injury, at 37.8%. Additionally, hospital admissions due to falls showed a continuous increase in the proportion of fall-related admissions among all hospitalizations, from 35.3% in 2013 to 49.7% in 2022 [2]. Falls are common among hospitalized patients. The fall rates per 1000 patient-days range from 2.4 in large tertiary university hospitals to 9.1 in geriatric hospital departments.

Such falls are caused by a combination of factors, including impaired balance [3], muscle weakness [4], declining vision [5,6], medication use [7], and environmental hazards. This is a major factor contributing to health deterioration [8,9], including fractures, head injuries, and complications, and can have particularly severe consequences for older adults. In fact, one study found that approximately 30% of inpatient falls result in injury, with 4% to 6% leading to serious outcomes such as fractures, subdural hematomas, excessive bleeding, or even death [10]. Injuries resulting from falls can lead to increased medical expenses, prolonged hospital stays, and economic losses due to potential medical malpractice litigation [11]. In addition to physical injuries, patients are disposed to mental harms such as anxiety, a fear of falling, and a loss of self-confidence [12]. Preventing falls in hospitals is a critical aspect of both patient safety and public health.

The occurrence of falls is frequently attributed to a multifactorial and interactive set of contributing factors. Furthermore, variations in the clinical environment and patient population characteristics may account for differences in the incidence of falls, injury severity, and situational factors surrounding falls. A study involving 1216 fall patients identified night shift hours, hospital room conditions, admission to the department of internal medicine, and being aged over 71 years as significant risk factors for falls [13]. Additional risk factors for falls identified in previous studies include disorientation, frequent urination, gait impairment, decreased endurance, and the amount of medication administered by caregivers within 72 h prior to the fall [14]. 

Although many studies have been conducted to effectively prevent and manage falls [15,16,17], further diverse research is needed to gain a comprehensive understanding of the causes and mechanisms of falls. In South Korea, previous studies have focused on identifying risk factors by comparing fall and non- fall groups [17,18,19,20], while relatively few have explored the characteristics of falls across different clinical departments. 

Given that nurses are primarily responsible for the care of patients who experience falls, examining the relationship between fall incidents and nursing shift schedules may offer valuable insights for developing more targeted and efficient fall prevention strategies.

In this study, we aimed to investigate the characteristics of inpatients at Kangwon National University Hospital (KNUH) who experienced falls, with these being classified into (1) surgical and internal medicine groups and (2) nursing shift hours to examine the characteristics of patients who experienced falls in each group. The objective was to identify the characteristics of inpatient falls, thereby contributing to the development of effective fall prevention strategies at the institutional level.

## 2. Materials and Methods

### 2.1. Study Design

This study was a retrospective, cross-sectional study that evaluated the characteristics of adult inpatients who experienced falls.

### 2.2. Study Population and Data Collection

The study included adult inpatients aged 18 years or older who experienced a fall during hospitalization between 1 January 2018 and 31 December 2023. Pediatric patients (<18 years), outpatients, and emergency department cases were excluded. The data were collected by the research team from electronic medical records and fall reports at a university hospital with more than 600 beds (KNUH). Among 83,208 adult inpatients during the study period, 495 fall cases were identified. To identify the factors influencing falls, variables such as demographics (age, sex, height, weight), main symptoms at admission, medications, comorbidities, alcohol history, smoking history, and Morse Fall Scale were analyzed for patients who experienced falls. 

Previous studies conducted in South Korea have reported higher fall incidence rates in departments such as rehabilitative medicine, neurosurgery, and orthopedic surgery [21], while some have indicated a higher incidence in internal medicine [13]. In this study, patients were categorized into internal medicine and surgical department groups (details of the departments can be found in Table 1). Among these, 336 internal medicine patients and 159 surgical patients experienced falls during their hospitalization. The time of fall occurrence was divided into day (7 a.m. to 3 p.m.), evening (3 p.m. to 11 p.m.), and night (11 p.m. to 7 a.m.) shifts based on nurses’ working hours.

Patients’ comorbidities were categorized as follows: cardiovascular disease (hypertension, stroke, heart failure, angina, myocardial infarction), neurological disease (epilepsy, Parkinson’s disease, dementia, delirium, psychotic disorders, syncope, depression), respiratory disease (pneumonia, chronic obstructive pulmonary disease, asthma, pleural effusion, bronchiectasis), malignancy, musculoskeletal disorders (osteoporosis, arthritis, fractures), digestive/urinary system diseases (including diabetes mellitus, chronic kidney disease, acute kidney injury, liver cirrhosis, and benign prostatic hyperplasia), and ocular disease (cataract). Malignancy cases were stratified by treatment status: under active treatment, treatment completed, or no treatment. The number of medications per patient was noted, with specific attention to high-risk drugs (e.g., sedatives, antidepressants, antipsychotics, diuretics, etc.). The duration of exposure to these drugs before a fall was also examined. The fall patients’ mental status (alert, confused, drowsy, stuporous) and activity status (independent, required assistance, bedridden) were also evaluated.

The Morse Fall Scale, which is routinely used in our institution for fall risk assessment among adult inpatients, was recorded both at admission and at the time of the fall. The Morse Fall Scale is one of the most used tools in fall prevention research and clinical trials [22]. The Morse falls score (MFS) assesses six variables (history of falling, secondary diagnosis, ambulatory aid, intravenous infusion, gait/transferring and mental status) to determine fall risk. Scores typically classify patients as low risk (0–24), moderate risk (25–44), and high risk (≥45), though thresholds may be adjusted locally [23]. According to the KNUH fall risk management policy, all admitted adult patients underwent fall risk screening using the Morse Fall Scale before 2020. Since 2020, patients aged ≥75 years have been considered a high-risk group for falls without separate evaluation, and fall prevention activities are immediately implemented. The dataset was approved by the Institutional Review Board of KNUH (IRB No. KNUH- 2023-12-007-001).

### 2.3. Statistical Analysis

In this study, chi-squared analysis was performed on categorical data, and a *t*-test and ANOVA were performed on continuous data to examine differences in the group distribution. All analyses were conducted with a *p*-value of 0.05, and SPSS version 29.0 was used.

## 3. Results

From 2018 to 2023, 83,208 internal medicine patients and 79,297 surgical patients were hospitalized at KNUH. Among these, 336 internal medicine patients and 159 surgical patients experienced falls during their hospitalization, accounting for 0.4% and 0.2%, respectively.

The length of stay for surgical patients was longer at 34.49 ± 47.52 d than that at 24.63 ± 28.37 d for internal medicine patients (*p*-value = 0.016). Falls occurred at a higher rate during night shifts (11 p.m. to 7 a.m.) in both internal medicine (40.2%) and surgical patients (37.1%), making it the highest among the three shifts. Cardiovascular, musculoskeletal, and eye diseases were more prevalent in surgical patients, whereas neurological and respiratory diseases were more common in internal medicine patients. The number of medications taken was higher in surgical patients (9.20 ± 4.328) than that in internal medicine patients (6.83 ± 4.013). Among fall risk medications, anticonvulsants were more commonly used in surgical patients, whereas diuretics were more frequently prescribed to internal medicine patients. Before the fall, narcotic analgesics were used for approximately 6.17 d in internal medicine patients and 3.77 d in surgical patients, indicating that falls occurred after a shorter duration of medication use in surgical patients. Additionally, a higher proportion of internal medicine patients than surgical patients could perform activities independently (Table 2).

When examining the characteristics based on nursing shifts, patients who experienced falls during night shifts were found to be older than those who fell during day or evening shifts. Furthermore, patients who fell during day shifts had a longer length of stay than those who fell during evening or night shifts. Among fall risk medications, laxative use was more prevalent during evening shifts. The duration of narcotic analgesic use before a fall was shorter during evening shifts than that during day or night shifts. Similarly, diuretics were administered for a shorter period before a fall during day shifts than that during evening or night shifts (Table 3).

## 4. Discussion

This study aimed to examine the incidence and characteristics of falls among adult patients aged 18 and older who were admitted to a university hospital with more than 600 beds (KNUH) over a six-year period from January 2018 to December 2023, and to identify the factors contributing to the risk of falls. The total number of patients who experienced falls was 336 out of 83,208 internal medicine patients and 159 out of 79,297 surgical patients, representing 0.4% and 0.2%, respectively. In patients who experienced falls, surgical patients had a longer length of stay than internal medicine patients. In both groups, the incidence rate was higher during night shifts.

According to a study conducted in South Korea, the fall incidence rate among patients aged ≥15 years in a tertiary general hospital over 1 year was reported to be 0.19%, equivalent to 1.9 falls per 1000 hospitalized patients [24]. In the United States, the large academic hospital fall rate was 3.1 falls per 1000 patient-days [25]. An evaluation of 6100 units in 1263 US hospitals over 27 months indicated a fall rate of 3.56 per 1000 bed days, with 26.1% of falls resulting in injury [26]. This study demonstrated a fall incidence rate similar to that shown in previous domestic studies.

One study found that the prevalence of inpatient falls was highest in the rehabilitation (1.915%) and internal medicine wards (1.181%), with the lowest rates observed in the orthopedic (0.145%) and rheumatology wards (0.213%) (*p* < 0.001). A majority of the falls (56.711%) occurred during late evening or night shifts [27].

Another study showed that the incidence of falls was the highest in the morning shift, followed by that in the night shift (24:00 to 08:00 h), and was the lowest during the evening shift (16:00 to 24:00 h) [28]. During the study period from January 2019 to December 2020, a total of 76 falls were documented, of which 18 occurred during the day shift and 58 during the night shift [29]. The higher incidence of falls during night shifts may be explained by reduced nurse staffing, while in daytime hours, fall prevention care may be deprioritized due to patient discharges and scheduled diagnostic procedures, leading to less time available for fall management compared to the evening shift [13].

Among the patients who experienced falls, cardiovascular disease, musculoskeletal disease, and eye disease were more prevalent in surgical patients. In contrast, neurological and respiratory diseases were more common in internal medicine patients. A domestic study showed that among the underlying conditions of patients who experienced falls, malignant neoplasms were the most common, affecting 192 patients (46.1%), followed by digestive and cardiovascular diseases [30]. Although it was not a study analyzing each department, one study that examined comorbidities among 1033 patients who experienced falls found that hypertension (73.7%), diabetes mellitus (36.3%), and congestive heart failure (23.6%) were the most commonly associated conditions [31].

Regarding medications, the average number of medications taken was higher in the surgical group (9.20 ± 4.328) than that in the internal medicine group (6.83 ± 4.013). Among fall risk medications, the use of anticonvulsants was higher in the surgical group than that in the internal medicine group, while diuretics were more commonly administered in the internal medicine group. The duration of narcotic analgesic use before the fall was approximately 6.17 d in the internal medicine group and 3.77 d in the surgical group, suggesting a shorter duration of medication use in the surgical group.

One study found that the medications taken by the group that experienced falls were as follows: narcotics (38.0%), antihypertensives (33.6%), bowel softeners (21.3%), and diuretics (19.7%) [18]. Another study demonstrated that medication exposure 24 h before a fall significantly increased the risk of inpatient falls in the four medication groups: benzodiazepine (odds ratio [OR] = 2.63, 95% confidence interval [CI] ¼ = 1.55–4.46), zolpidem (OR = 2.38, 95% CI = 1.04–5.43), narcotics (OR = 2.13, 95% CI = 1.16–3.94), and antihistamines (OR = 3.00, 95%CI = 1.19–7.56) [32].

When examining the characteristics based on nurses’ working hours, it was found that patients who experienced falls during night shifts were older than those who fell during day or evening shifts. Additionally, patients who experienced falls during day shifts had a longer length of stay than those who fell during evening or night shifts. Marcin et al. reported that falls occurred most frequently between 24:00 and 6:00 and were more prevalent in female patients than that in male patients [27]. Another study showed that the occurrence of falls was higher during the night shift (46%) than that during either the morning (30%) or afternoon (24%) shifts [33].

This study examined the various characteristics of adult patients who experienced falls over a 6-year period; however, it had the following limitations.

First, we used data from just one hospital, resulting in a small study population. 

Second, this study was designed as a cross-sectional study targeting patients who experienced falls over a 6-year period. Therefore, to investigate the correlation with falls, future comparative studies with a control group will be necessary. Nonetheless, the strength of this study lies in its analysis of hospitalized patients, taking into account the various variables that influence falls.

Falls occurring during hospitalization are influenced by a wide range of factors, including patient-related, disease-related, nursing-care-related, and environmental elements. This study is meaningful in that it classified patients who experienced falls according to medical or surgical departments and nursing shift hours, aiming to identify various influencing variables and analyze their characteristics. Such research can serve as a fundamental resource for understanding the characteristics of patients who experience falls and for developing effective fall prevention strategies.

## 5. Conclusions

Falls are closely related to patient safety issues. In this study, we examined the characteristics of patients who experienced falls, categorizing them by internal medicine and surgical departments, as well as by nursing shift hours. The incidence of falls was 0.4% (336/83,208) of internal medicine patients and 0.2% (159/79,297) of surgical patients, with the highest rate during night shifts (internal medicine 40.2%, surgical 37.1%) at KNUH. Surgical patients had longer hospital stays (34.49 ± 47.52 vs. 24.63 ± 28.37 days), took more medications (9.20 vs. 6.83), and experienced falls sooner after narcotic use (3.77 vs. 6.17 days) than internal medicine patients. Patients who experienced falls during night shifts were older than those who fell during day or evening shifts (70.56 (±13.976) vs. 65.58 (±15.029) vs. 65.79 (±14.908) years). Additionally, those who fell during day shifts had a longer length of stay than those who fell during other shifts (33.27 (±49.306) vs. 27.92 (±28.496) vs. 23.36 (±26.731) days). Although we examined the characteristics of high-risk medications for falls, no significant differences were observed.

Falls are preventable medical incidents, and further research is needed to enhance the reliability of the findings. A large-scale population-based cohort study design is essential, and considering the various factors influencing falls, a meta-analysis integrating studies conducted in diverse settings is warranted.

## Figures and Tables

**Table 1 ijerph-22-00748-t001:** Details of the departments for each group.

Groups	Departments
Internal Medicine	Cardiology
	Pulmonology
	Gastro enterology
	Nephrology
	Endocrinology
	Hemato-oncology
	Infectious diseases
	Allergy
	Rhematology
	Neurology
	Rehabilitation
	Psychiatry
	Geriatrics
	Emergency
Surgery	General Surgery
	Orthopedic Surgery
	Neuro-Surgery
	Thoracic Surgery
	Otorhinolaryngology
	Plastic Surgery
	Urology
	Dental Surgery
	Obstetrics and Gynaecology
	Ophthalmology

**Table 2 ijerph-22-00748-t002:** Patients classified by internal medicine and surgery.

Variables	Frequency or Mean (±SD)	Internal Medicine	Surgical	Chi-Square or T (*p*-Value)
Demographics				
Sex, male, *n* (%)				
Age, mean (±SD)	67.59 (±14.752)	67.53 (±14.352)	67.72 (±15.610)	−0.130 (0.897)
Height cm, mean (±SD)	161.807 (±9.349)	162.04 (±9.261)	161.31 (±9.545)	0.806 (0.421)
Weight kg, mean (±SD)	60.70 (±12.184)	59.43 (±11.317)	63.39 (±13.488)	−3.411 (0.001)
BMI kg/m^2^, mean (±SD)	23.14 (±4.01)	22.61 (±3.816)	24.27 (±4.199)	−4.353 (<0.001)
Length of stay day, mean (±SD)	27.80 (±35.905)	24.63 (±28.367)	34.49 (±47.520)	−2.421 (0.016)
Time of falls				
Day shift (7 a.m.–3 p.m.)	154 (31.1)	108 (32.1)	46 (28.9)	2.053 (0.358)
Evening shift (3 p.m.–11 p.m.)	147 (29.7)	93 (27.7)	54 (34.0)	
Night shift (11 p.m.–7 a.m.)	194 (39.2)	135 (40.2)	59 (37.1)	
Underlying disease, n (%)				
Cardiovascular disease	275 (55.6)	171 (50.9)	104 (65.4)	9.210 (0.002)
Neurological disease	80 (16.2)	63 (18.8)	17 (10.7)	5.172 (0.023)
Respiratory disease	82 (16.6)	67 (19.9)	15 (9.4)	8.620 (0.003)
Malignancy, *n* (%)				
In treatment	87 (17.6)	74 (22.0)	13 (8.2)	14.487 (0.001)
End of treatment	34 (6.9)	23 (6.8)	11 (6.9)	
No treatment	374 (75.6)	239 (71.1)	135 (84.9)	
Musculoskeletal disease, *n* (%)	89 (18.0)	48 (14.3)	41 (25.8)	9.680 (0.002)
Digestive/urinary system disease, *n* (%)	236 (47.7)	165 (49.1)	71 (44.7)	0.858 (0.354)
Eye disease (Cataract), *n* (%)	33 (6.7)	16 (4.8)	17 (10.7)	6.099 (0.014)
Alcohol, *n* (%)				
Present	102 (20.6)	65 (19.3)	37 (23.3)	9.823 (0.007)
Past	77 (15.6)	64 (19.0)	13 (8.2)	
Not applicable	316 (63.8)	207 (61.6)	109 (68.6)	
Smoking, *n* (%)				
Present	121 (24.4)	79 (23.5)	42 (26.4)	9.136 (0.010)
Past	87 (17.6)	71 (21.1)	16 (10.1)	
Not applicable	287 (58.0)	186 (55.4)	101 (63.5)	
Number of medications taken, mean (±SD)	7.56 (±4.252)	6.83 (±4.013)	9.20 (±4.328)	−5.606 (<0.001)
Fall risk medications, *n* (%)				
Sleep sedative/psychotropic drug	43 (8.7)	28 (8.3)	15 (9.4)	0.165 (0.685)
Antidepressants	42 (8.5)	28 (8.3)	14 (8.8)	0.031 (0.860)
Anxiolytics	83 (16.8)	61 (18.2)	22 (13.8)	1.442 (0.230)
Antipsychotics	118 (23.8)	85 (25.3)	33 (20.8)	1.227 (0.268)
Narcotic analgesics	95 (19.2)	64 (19.0)	31 (19.5)	0.014 (0.906)
Anticonsulsant	79 (16.0)	42 (12.5)	37 (23.3)	9.334 (0.002)
Diuretic	75 (15.2)	60 (17.9)	15 (9.4)	5.956 (0.015)
Laxative	123 (24.8)	81 (24.1)	42 (26.4)	0.308 (0.579)
Duration of fall-risk medication use up to the time of the fall occurrence, *n* (%)				
Sleep sedative/psychotropic drug	7.33 (±12.727)	5.39 (±5.659)	10.93 (±20.069)	−1.375 (0.088)
Antidepressants	6.88 (±6.444)	6.46 (±5.997)	7.71 (±7.426)	−0.588 (0.560)
Anxiolytics	7.14 (±6.964)	6.08 (±5.838)	10.09 (±8.922)	−1.961 (0.060)
Antipsychotics	8.51 (±8.731)	8.27 (±7.921)	9.12 (±10.653)	−0.473 (0.637)
Narcotic analgesics	5.39 (±5.788)	6.17 (±6.558)	3.77 (±3.263)	2.379 (0.019)
Anticonsulsant	11.38 (±13.058)	11.33 (±13.935)	11.43 (±12.178)	−0.033 (0.973)
Diuretic	8.89 (±12.081)	7.62 (±9.655)	14.00 (±18.540)	−1.290 (0.215)
Laxative	8.93 (±9.928)	9.28 (±10.228)	8.26 (±9.407)	0.540 (0.590)
Mental status (Not included semi-coma), *n* (%)				
Alert	420 (84.8)	284 (84.5)	136 (85.5)	0.086 (0.770)
Confuse	61 (12.3)	42 (12.5)	19 (11.9)	0.030 (0.862)
Drowsy	8 (1.6)	6 (1.8)	2 (1.3)	0.189 (0.664)
Stupor	1 (0.2)	0 (0.0)	1 (0.6)	2.117 (0.146)
Activity status, *n* (%)				
Independent	181 (36.6)	146 (43.5)	35 (22.0)	21.388 (<0.001)
Required help	266 (53.7)	156 (46.4)	110 (69.2)	22.447 (<0.001)
Bed ridden status	43 (8.7)	30 (8.9)	13 (8.2)	0.077 (0.781)
Morse fall scale, mean (±SD)				
The time of admission	28.27 (±17.451)	29.34 (±16.585)	25.85 (±19.096)	1.848 (0.066)
The time of fall occurrence	54.49 (±18.147)	53.85 (±18.229)	55.85 (±17.954)	−1.144 (0.253)
Use of the restraint band, *n* (%)				
Yes	14 (2.8)	11 (3.3)	3 (1.9)	
No	481 (97.2)	325 (96.7)	156 (98.1)	0.755 (0.385)

**Table 3 ijerph-22-00748-t003:** Patients classified by nurses’ working time.

Variables	Frequency or Mean (±SD)	Day Shift (7 a.m.–3 p.m.)	Evening Shift (3 p.m.–11 p.m.)	Night Shift (11 p.m.–7 a.m.)	Chi-Square or T (*p*-Value)
Demographics					
Sex, male, *n* (%)					
Age, mean (±SD)	67.59 (±14.752)	65.58 (±15.029)	65.79 (±14.908)	70.56 (±13.976)	6.597 (0.001)
Height cm, mean (±SD)	161.807 (±9.349)	162.00 (±9.260)	161.52 (±9.079)	161.87 (±9.658)	0.107 (0.899)
Weight kg, mean (±SD)	60.70 (±12.184)	60.83 (±12.072)	61.22 (±13.072)	60.21 (±11.607)	0.297 (0.743)
BMI kg/m^2^, mean (±SD)	23.14 (±4.010)	23.16 (±4.072)	23.35 (±4.049)	22.97 (±3.952)	0.371 (0.690)
Length of stay day, mean (±SD)	27.80 (±35.905)	33.27 (±49.306)	27.92 (±28.496)	23.36 (±26.731)	3.299 (0.038)
Department					
Internal medicine	336 (67.9)	135 (69.6)	93 (63.3)	108 (70.1)	2.053 (0.358)
Surgery	159 (32.1)	59 (30.4)	54 (36.7)	46 (29.9)	
Underlying disease					
Cardiovascular disease	275 (55.6)	88 (57.1)	81 (55.1)	106 (54.6)	0.235 (0.889)
Neurological disease	80 (16.2)	19 (12.3)	24 (16.3)	37 (19.1)	2.878 (0.237)
Respiratory disease	82 (16.6)	24 (15.6)	18 (12.2)	40 (20.6)	4.398 (0.111)
Malignancy					
In treatment	87 (17.6)	37 (19.1)	24 (16.3)	26 (16.9)	4.072 (0.396)
End of treatment	34 (6.9)	17 (8.8)	11 (7.5)	6 (3.9)	
No treatment	374 (75.6)	140 (72.2)	112 (76.2)	122 (79.2)	
Musculoskeletal disease	89 (18.0)	26 (16.9)	26 (17.7)	37 (19.1)	0.291 (0.865)
Digestive/urinary system disease	236 (47.7)	76 (49.4)	65 (44.2)	95 (49.0)	1.008 (0.604)
Eye disease (Cataract)	33 (6.7)	20 (10.3)	4 (2.7)	9 (5.8)	7.982 (0.018)
Alcohol					
Present	102 (20.6)	34 (17.5)	37 (25.2)	31 (20.1)	6.711 (0.152)
Past	77 (15.6)	37 (19.1)	15 (10.2)	25 (16.2)	
Not applicable	316 (63.8)	123 (63.4)	95 (64.6)	98 (63.3)	
Smoking					
Present	121 (24.4)	36 (18.6)	44 (29.9)	41 (26.6)	7.652 (0.105)
Past	87 (17.6)	40 (20.6)	20 (13.6)	27 (17.5)	
Not applicable	287 (58.0)	118 (60.8)	83 (56.5)	86 (55.8)	
Number of medications taken	7.56 (±4.252)	7.60 (±4.075)	7.43 (±4.449)	7.62 (±4.261)	0.085 (0.918)
Fall risk medications					
Sleep sedative/psychotropic drug	43 (8.7)	12 (7.8)	12 (8.2)	19 (9.8)	0.506 (0.777)
Antidepressants	42 (8.5)	14 (7.2)	12 (8.2)	16 (10.4)	1.141 (0.565)
Anxiolytics	83 (16.8)	31 (16.0)	25 (17.0)	27 (17.5)	0.157 (0.925)
Antipsychotics	118 (23.8)	52 (26.8)	36 (24.5)	30 (19.5)	2.585 (0.275)
Narcotic analgesics	95 (19.2)	46 (23.7)	25 (17.0)	24 (15.6)	4.300 (0.116)
Anticonsulsant	79 (16.0)	33 (17.0)	29 (19.7)	17 (11.0)	4.496 (0.106)
Diuretic	75 (15.2)	24 (12.4)	22 (15.0)	29 (18.8)	2.792 (0.248)
Laxative	123 (24.8)	46 (23.7)	50 (34.0)	27 (17.5)	11.161 (0.004)
Duration of fall-risk medication use up to the time of the fall occurrence					
Sleep sedative/psychotropic drug	7.33 (±12.727)	4.92 (±5.195)	14.58 (±21.483)	4.26 (±5.526)	1.295 (0.295)
Antidepressants	6.88 (±6.444)	5.69 (±4.094)	7.33 (±7.947)	7.86 (±7.430)	0.452 (0.640)
Anxiolytics	7.14 (±6.964)	6.96 (±5.619)	6.48 (±7.200)	7.84 (±7.925)	0.272 (0.762)
Antipsychotics	8.51 (±8.731)	6.90 (±5.604)	10.14 (±9.940)	8.31 (±9.262)	1.153 (0.319)
Narcotic analgesics	5.39 (±5.788)	7.25 (±6.271)	3.44 (±3.969)	5.48 (±6.124)	2.764 (0.068)
Anticonsulsant	11.38 (±13.058)	10.53 (±11.912)	13.00 (±13.792)	10.39 (±13.210)	0.347 (0.708)
Diuretic	8.89 (±12.081)	5.97 (±4.508)	8.14 (±8.351)	13.13 (±18.739)	2.460 (0.093)
Laxative	8.93 (±9.928)	8.48 (±8.881)	8.78 (±10.574)	9.37 (±9.979)	0.077 (0.926)
Mental status (Not included semi-coma)					
Alert	420 (84.8)	159 (82.0)	128 (87.1)	133 (86.4)	2.102 (0.350)
Confuse	61 (12.3)	30 (15.5)	14 (9.5)	17 (11.0)	3.072 (0.215)
Drowsy	8 (1.6)	2 (1.0)	4 (2.7)	2 (1.3)	1.644 (0.440)
Stupor	1 (0.2)	1 (0.5)	0 (0.0)	0 (0.0)	1.555 (0.460)
Activity status					
Independent	181 (36.6)	66 (34.0)	54 (36.7)	61 (39.6)	1.159 (0.560)
Required help	266 (53.7)	106 (54.6)	79 (53.7)	81 (52.6)	0.144 (0.931)
Bed ridden status	43 (8.7)	20 (10.3)	13 (8.8)	10 (6.5)	1.582 (0.453)
Morse fall scale					
The time of admission	28.27 (±17.451)	28.33 (±17.542)	27.28 (±17.981)	28.93 (±17.046)	0.327 (0.721)
The time of fall occurrence	54.49 (±18.147)	51.27 (±17.524)	53.32 (±16.808)	57.91 (±19.089)	6.286 (0.002)
Use of the restraint band					
Yes	14 (2.8)	5 (2.6)	5 (3.4)	4 (2.6)	
No	481 (97.2)	189 (97.4)	142 (96.6)	150 (97.4)	0.250 (0.883)

## Data Availability

The data presented in this study are available upon request from the corresponding author due to privacy and ethical restriction.

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
