# Peer review of "Evaluation of Risk Factors for Fall Incidence Based on Statistical Analysis"

_ijerph, 2025, doi:10.3390/ijerph22050748_

Round 1
Reviewer 1 Report
Comments and Suggestions for Authors
Article
Evaluation of Risk Factors for Fall Incidence Based on Statistical Analysis
The topic of the manuscript is interesting. The statistical methods are chosen correctly.
I recommend a structured Abstract according to the model: Introduction. Aim/Objective. Methods. Results. Conclusions. I would add two key words.
The article lacks a detailed description of the research sample and demographic data. This is a cross-sectional study, authors undefined variables.
I propose dividing table No. 2 and table No. 3 from a formal point of view.
The ethical aspect was respected. I miss the detailed study design, description of contributing factors of falls in specific patients, etc.
The limitations of the study are described at the end of the discussion. The study was designed as a cross-sectional study focusing on patients who experienced falls over a 6-year period.
The conclusion evaluates the results of the study, I would add numerical results.
The last sentence is of a general nature: Falls are preventable medical incidents, and further research is needed to explore the characteristics, risk factors, and fall assessments. Suggest specific ways
The article uses old literary sources, for example: 2007, 2008, 1995.
Accept after recommended revisions.
Reviewer 2 Report
Comments and Suggestions for Authors
The article requires further refinement and clarification. Please see my comments below…
INTRODUCTION
The introduction section is underdeveloped. While it mentions the need for further diverse research, this point is not clearly explained or supported. Additionally, the aim of the study and its relevance are not clear.
METHODS
Methods sections is also quite poor…
Please define inclusion and exclusion criteria and their justification…
It is not clear whether the falls occurred before or while at the hospital...
Why were the patients categorized into internal medicine and surgical department groups?
How and by whom were the data collected for the study?
Please describe the Morse Fall Scale and the pertinence for its use…
RESULTS
In tables, some parameters need units…
DISCUSSION
I do not understand what is discussed in the first paragraph…
There is a lack of discussion on the implications for practice based on the data obtained.
Round 2
Reviewer 2 Report
Comments and Suggestions for Authors
None.